# Stromal and epithelial transcriptional map of initiation progression and metastatic potential of human prostate cancer

Svitlana Tyekucheva[1,2], Michaela Bowden[3], Clyde Bango[4], Francesca Giunchi[5], Ying Huang[4],
Chensheng Zhou [3], Arrigo Bondi[6], Rosina Lis[3,7], Mieke Van Hemelrijck[8], Ove Andrén[9], Sven-Olof Andersson[9],
R. William Watson[10], Stephen Pennington[10], Stephen P. Finn[11], Neil E. Martin[12], Meir J. Stampfer[13,14,15],
Giovanni Parmigiani[1,2], Kathryn L. Penney[13,14], Michelangelo Fiorentino[5], Lorelei A. Mucci[13,14] &
Massimo Loda[4,7,16]

While progression from normal prostatic epithelium to invasive cancer is driven by molecular alterations, tumor cells and cells in the cancer microenvironment are co-dependent and co-evolve. Few human studies to date have focused on stroma. Here, we performed gene expression profiling of laser capture microdissected normal non-neoplastic prostate epithelial tissue and compared it to non-transformed and neoplastic low-grade and high-grade prostate epithelial tissue from radical prostatectomies, each with its immediately surrounding stroma. Whereas benign epithelium in prostates with and without tumor were similar in gene expression space, stroma away from tumor was significantly different from that in prostates without cancer. A stromal gene signature reflecting bone remodeling and immune-related pathways was upregulated in high compared to low-Gleason grade cases. In validation data, the signature discriminated cases that developed metastasis from those that did not. These data suggest that the microenvironment may influence prostate cancer initiation, maintenance, and metastatic progression.

[1] Department of Biostatistics and Computational Biology, Dana-Farber Cancer Institute, 450 Brookline Ave, Boston, MA 02215, USA. [2] Department of Biostatistics, Harvard T.H. Chan School of Public Health, 677 Huntington Ave, Boston, MA 02115, USA. [3] Department of Medical Oncology, Dana-Farber Cancer Institute, 450 Brookline Ave, Boston, MA 02215, USA. [4] Department of Oncologic Pathology, Dana-Farber Cancer Institute, 450 Brookline Ave, Boston, MA 02215, USA. [5] Department of Pathology, Addarii Institute of Oncology, S.Orsola-Malpighi Teaching Hospital, University of Bologna, Viale Ercolani 4/2, 40138 Bologna, Italy. [6] Department of Surgical Pathology, Maggiore Hospital, Largo Nigrisoli 2, 40133 Bologna, Italy. [7] Department of Pathology, Brigham and Women's Hospital, Harvard Medical School, 75 Francis St, Boston, MA 02115, USA. [8] King's College London, Division of Cancer Studies, Translational Oncology & Urology Research, Guy's Hospital, London SE1 9RT, UK. [9] Department of Urology, School of Health and Medical Sciences, Örebro University Hospital, Örebro SE 701 85, Sweden. [10] School of Medicine, UCD Conway Institute of Biomolecular and Biomedical Research, University College Dublin, Belfield, Dublin 4, Ireland. [11] Department of Histopathology and Morbid Anatomy, School of Medicine, Trinity College Dublin, Dublin, Ireland. [12] Department of Radiation Oncology, Dana-Farber Cancer Institute, Brigham and Women's Hospital, Harvard Medical School, 75 Francis St, Boston, MA 02115, USA. [13] Channing Division of Network Medicine, Department of Medicine, Brigham and Women's Hospital and Harvard Medical School, 181 Longwood Ave, Boston, MA 02115, USA. [14] Department of Epidemiology, Harvard T.H. Chan School of Public Health, 677 Huntington Ave, Boston, MA 02115, USA. [15] Department of Nutrition, Harvard T.H. Chan School of Public Health, Boston, MA 02215, USA. [16] The Broad Institute, 415 Main St, Cambridge, MA 02142, USA. Svitlana Tyekucheva and Michaela Bowden contributed equally to this work. Correspondence and requests for materials should be addressed to M.L. (email: massimo_loda@dfci.harvard.edu)

The prostate consists of the glandular epithelium and supporting stroma. This connective stroma is comprised of fibroblasts, myofibroblasts, smooth muscle cells, vascular endothelial cells, nerve cells, and inflammatory cells. While prostate cancer arises from the epithelial component of the gland, the surrounding stroma is increasingly recognized as an important contributor in the process of carcinogenesis[1, 2] and a driver of cancer progression. Experimental models demonstrate that altered stromal cells can induce tumor formation in non-cancerous prostate epithelial cells[2] and in cell lines derived from prostate cancer[3]. Benign prostate epithelial cells are more proliferative and ultimately undergo transformation when combined with prostate cancer-derived fibroblasts[2, 4]. It is also clear that the stroma can morphologically and functionally change in the presence of cancer and other insults. Compared to normal stroma, there is a switching of the cellular phenotype[5], remodeling of the extracellular matrix[6] increases in expression of growth factors and proteases[7] increased angiogenesis[8], and change in inflammatory cells[9]. The bidirectional signaling between epithelial cells and stromal constituents during normal prostate homeostasis is disrupted early in tumorigenesis (reviewed in ref. [10]). The consequences are diverse and range from deposition of extracellular matrix, to recruitment of inflammatory cells, production of miRNA, promotion of tissue regeneration and angiogenesis, ultimately resulting in stimulation of growth and survival of tumor cells[11–13]. When the stromal compartment becomes reactive, normal fibroblasts are replaced by cancer-associated fibroblasts (CAFs). The increase of CAFs, which begins around in situ lesions, evolves during prostate tumorigenesis and is inversely proportional to tumor differentiation[14].

Signaling factors from the microenvironment influence epithelial cells to acquire properties such as increased motility, proliferation or migratory and invasive behavior. To this end, TGFβ and Wnt signaling pathways have been shown to play important regulatory roles in stromal-epithelial interactions in both prostate development and tumorigenesis[10, 15, 16]. A variety of additional growth factors produced by stromal cells affect tumor cell survival[17]. In addition, soluble cytokine and chemokines influence the interaction between the epithelial and stromal compartments during prostate cancer progression. For example, peri-prostatic adipose tissue can affect migration of prostate cancer cells via secretion of CCL7 by adipocytes[18]. Finally, androgen receptor, expressed by a subset of myofibroblasts in the prostate stroma, may regulate the expression of growth factors secreted by these cells[19]. Thus, tumor growth and biologic behavior is strongly regulated by the extracellular milieu.

Most human studies have focused on the mutational landscapes in tumors in an attempt to predict biologic and clinical behavior of human prostate cancer[20, 21]. In addition, epigenetic and transcriptional epithelial signatures are associated with the degree of differentiation and are an important adjunct in predicting aggressive and indolent behavior[22–24]. While these contribute to additional independent prognostic information, they could be further improved by knowledge of the contribution of stromal elements. While it has been recently shown that the stroma adjacent to prostate cancer epithelium does not harbor clonal DNA alterations and appears to be genetically stable[25], biological behavior of the epithelial component of the tumor, may be affected by variability of gene expression in the stroma. In turn, epithelial alterations may condition stromal behavior. For instance, hyperactivated focal adhesion kinase (FAK) activity has been shown to be an important regulator of the fibrotic and immunosuppressive stromal microenvironment in pancreatic cancer[26]. Additionally, stromal gene expression signatures predict outcome in breast[27–29] and colorectal[30] cancer patients.

Laser capture microdissection (LCM) has facilitated the isolation and study of specific cellular populations within the prostate tumor microenvironment. This labor-intensive technology, however, limits large-scale studies. To date, differences between the tumor and its adjacent stroma in prostate cancer[31] between normal and reactive stroma[32], and differences between benign and tumor epithelium[32–34] have been addressed utilizing LCM, albeit on a small scale. Prior analyses were centered predominantly on the epithelial compartment. Limited studies of stromal gene expression using high-throughput assays exist for prostate cancer aggressiveness. One such study showed alterations in neurogenesis, axonogenesis, and DNA damage/repair pathway to be associated with grade 3 reactive stroma[32].

Here, we hypothesize that progression of normal prostate to prostatic intraepithelial neoplasia (PIN) to invasive cancer is driven by molecular alterations in both epithelium and stroma, and that changes in the microenvironment can potentially contribute to tumor initiation, maintenance and progression. We find that gene expression of non-transformed epithelial and stromal tissues differ in prostates with and without tumor, and how the stromal genes are associated with prostate cancer progression and aggressiveness.

## Results

**Experimental design.** We performed gene expression profiling of laser capture microdissected tissue specimens from 12 low-grade (Gleason 3 + 3) and 13 high-grade (Gleason 8 and higher) radical prostatectomy (RP) and 5 cystoprostatectomy cases. For each RP case, we took 6 regions of interest: tumor (T), PIN (P) and benign (B) epithelium each with its adjacent stroma (sT, sP, sB).

| Table 1 Clinical characteristics of the LCM cohort | |
| --- | --- |
| **Total number of cases** | **30** |
| Mean (s.d.) age at diagnosis | 63.7 (7.7) |
| *Clinical stage* | |
| T1 | 4 |
| T1c | 1 |
| T2 | 12 |
| T2a | 2 |
| T3 | 1 |
| M1 | 1 |
| NA | 7 |
| | |
| *Pathological stage* | |
| pT1 | 2 |
| pT2 | 4 |
| pT2a | 2 |
| pT2b | 1 |
| pT2c | 7 |
| pT3a | 4 |
| pT3b | 4 |
| M1 | 1 |
| NA | 5 |
| | |
| *Gleason score* | |
| 3 + 3 | 12 |
| ≥8 | 13 |
| | |
| *Tissue type* | |
| RP | 25 |
| Cystoprostatectomy | 5 |

*RP* radical prostatectomy, *s.d.* standard deviation

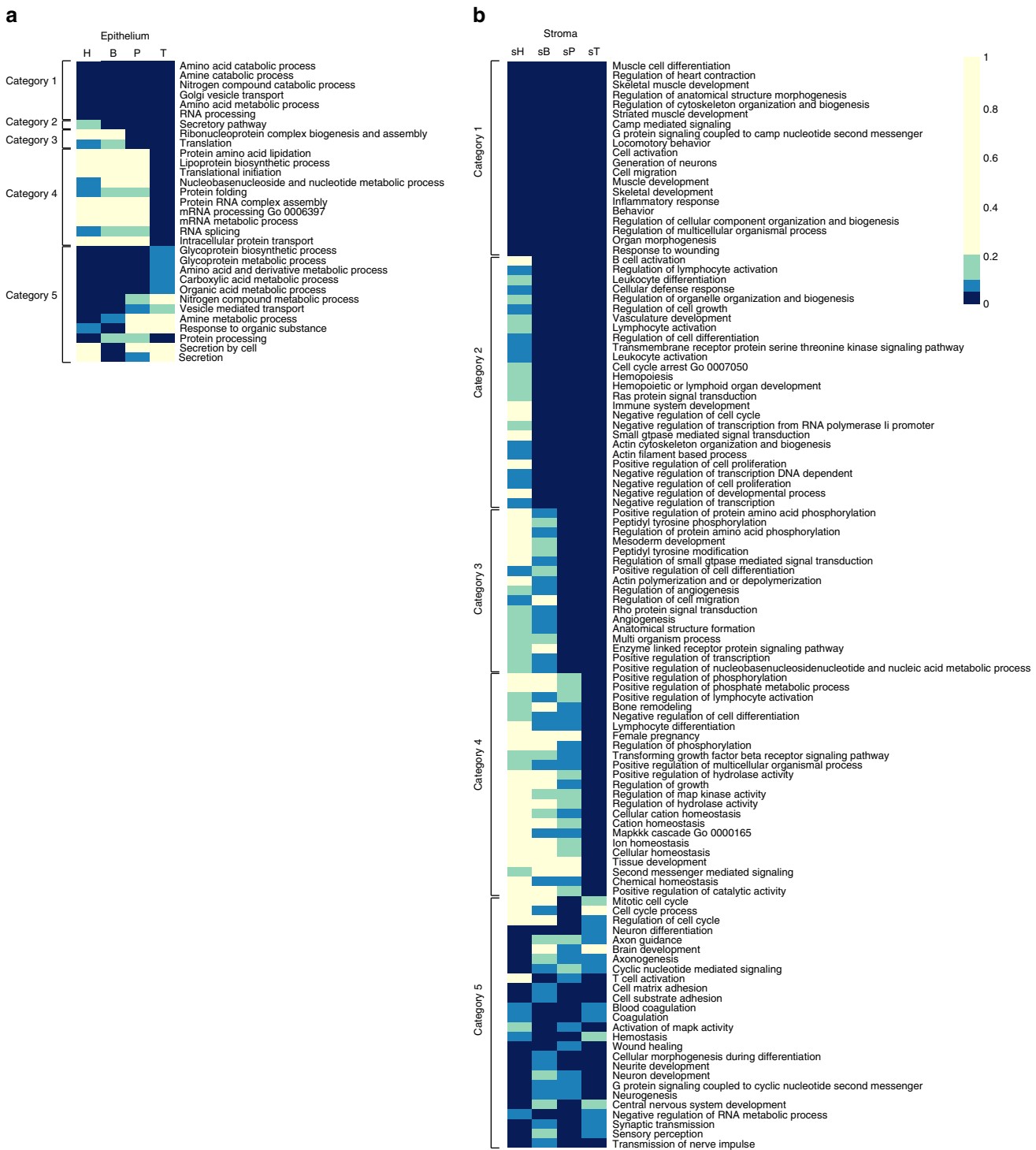

**Fig. 1** Heatmaps of the GO biological processes enriched in **a** epithelial and **b** stromal compartments across healthy prostate tissues and stages of prostate cancer progression. The cells in the heatmaps are colored according to the FDR of the process in the gene set analysis. *Dark blue* color corresponds to significance at 0.05 level and *yellow* to FDR > 0.2. Categories across compartments show conserved to unique processes from H to B to P to T. Most relevant pathways are summarized in categories: Category 1 (*top*) epithelial: amino acid metabolism; Category 2 epithelial: secretory pathway; Category 3 epithelial: RNA synthesis; Category 4 epithelial: RNA, protein, and lipid synthesis; Category 5 epithelial: miscellaneous; Category 1 (*top*) stromal: muscle development and localization; Category 2 stromal: immune regulation, angiogenesis and cell proliferation; Category 3 stromal: signal transduction, cell migration and angiogenesis; Category 4 stromal: TGF beta, signal transduction and bone remodeling; Category 5 stromal: miscellaneous

For cystoprostatectomy we studied benign epithelium and adjacent stroma (H.B and H.sB). Cystoprostatectomies were confirmed not to harbor prostate cancer foci through review of the entire submitted specimen. Clinicopathological features of the cohort are described in Table 1.

**Differences between compartments across progression.** As expected from our experimental design, the major share of variability in gene expression was explained by differences between epithelial and stromal tissue compartments (Supplementary Fig. 1A). Many of the differentially expressed

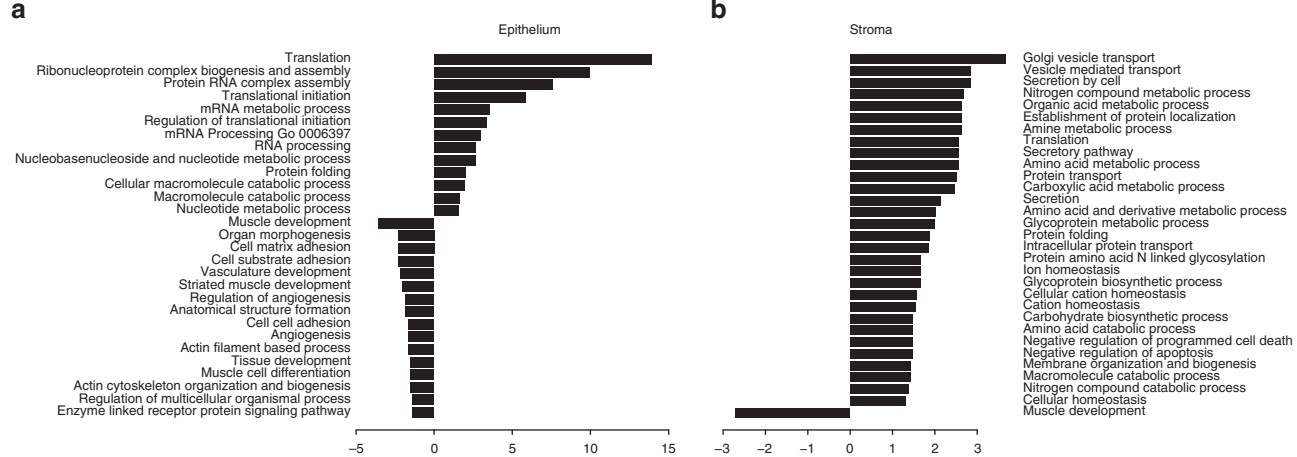

**Fig. 2** GO biological processes differentially enriched between **a** Benign and tumor epithelium; **b** Benign and tumor adjacent stroma. Lengths of the bars are equal to −log10(FDR) values from the gene set analysis. *Negative values* indicate enrichment in benign epithelium or stroma, respectively, *positive values* indicate enrichment in the tumor or tumor adjacent stroma

genes as well as pathways were shared across the H.B-H.sB, B-sB, P-sP and T-sT comparisons (Supplementary Fig. 1B and Fig. 1). The GO biological processes commonly upregulated in the epithelium, and which were maintained through "progression" to invasive tumors, included amino acid metabolism, RNA processing, protein translation and post-translational modification (Fig. 1a).

Common processes upregulated in stroma were mostly comprised of muscle development as well as changes in cytoskeletal structure (Fig. 1b). Among processes upregulated in all stromal components of the RP specimens, we find increasing occurrence of immune-related pathways, such as lymphocyte differentiation and activation. Interestingly, the bone remodeling pathway was upregulated more strongly in the stroma adjacent to the tumors.

**Differences within compartments across progression**. As proof of principle examples, *TP63*, a marker of normal basal cells of the prostate gland was upregulated in benign microdissected epithelial samples compared to invasive cancer, while *AMACR* and *ERG* were all upregulated in the tumor microdissected epithelial samples compared to benign epithelium and, to a lesser extent, PIN (Supplementary Fig. 2).

Gene set analysis in tumor epithelium showed pathways associated with nucleotide metabolism, translation, and RNA processing (Fig. 2a). Translation, protein folding, as well as negative regulation of apoptosis were upregulated in the tumor adjacent stroma (Fig. 2b). Muscle development GO biological process decreased in tumor-associated stroma, consistent with the transformation of stromal composition from mainly muscle cells to myofibroblasts and fibroblasts.

**Differences between RP and cystoprostatectomy cases**. We compared the benign epithelial glands from the cystoprostatectomy and RP (B-H.B) and found 15 differentially expressed probesets (moderated *t*-tests, FDR < 0.05, FC≥1.5; Fig. 3a). In the comparisons of the adjacent stroma from the cystoprostatectomy and RP tissues (sB-HsB), a larger number of probesets (*n* = 130; Fig. 3b, c) were statistically significant. Forty-two of them were probesets corresponding to small nucleolar mRNAs (almost all C/ D box), all overexpressed in normal stroma from RP specimens. The GO biological processes associated with the sB-HsB differentially expressed genes included N-linked glycosylation,

membrane and Golgi transport, and the unfolded protein response (Fig. 3d).

Interestingly, the hierarchical clustering revealed greater similarity in the expression of stromal genes between stroma adjacent to benign epithelium in the prostates with no tumor (cystoprostatectomies) and the benign stroma from prostates with high-grade tumors. Similar effect is observed using principal component analysis (PCA) projection of all genes (Supplementary Fig. 3), even though the physical distance between sB regions selected for analysis and the closest tumor focus, on average was smaller for high-grade cases (*t*-test; *P* = 0.04). This might suggest, that stroma surrounding Gleason 3 + 3 cases is inherently different. In the direct comparisons of the sB from high-grade and low-grade cases no genes reached statistical significance.

**Differences between high-grade and low-grade tumors**. Gleason grade is one of the strongest clinical predictors of prostate cancer progression and outcomes. We identified genes differentially expressed between high-grade and low-grade epithelium (T.high-T.low) and in adjacent stroma (sT.high-sT.low). A TGF-β-responsive marker and functional regulator of prostate cancer metastasis to bone, *ALCAM* (FDR = 0.005)[35] was identified as the only significantly differentially expressed gene in the epithelium comparison. Differences between gene expression in the sT.high-sT.low comparison, however, were more striking with 27 probesets corresponding to 24 unique gene symbols were differentially expressed in stroma (Table 2). All genes were upregulated in high-Gleason grade cases. The genes comprising this stromal signature include a group of genes overexpressed in osteoblasts and osteoblast-like cells, as well as some gene overexpressed in macrophages, T and B cells, even though these cells were scant (Supplementary Data 1 and Supplementary Fig. 4). GO biological processes involved in immune response as well as complement activation and skeletal system development were significantly enriched in our signature (Supplementary Data 2). This signature features wound healing and metastasis markers (*SFRP2*, *SFRP4*, *THBS2*), hematopoietic bone marrow markers (SULF1, COL1A1), immune cell markers (*HLA-DRB1*, *FCGR2C*), and complement cascade genes (*C1S*, *C1QA*, *C1QB*, *C1QC*).

The single sample gene set enrichment (ssGSEA[36]) score was used to summarize expression of these 27 probesets in the

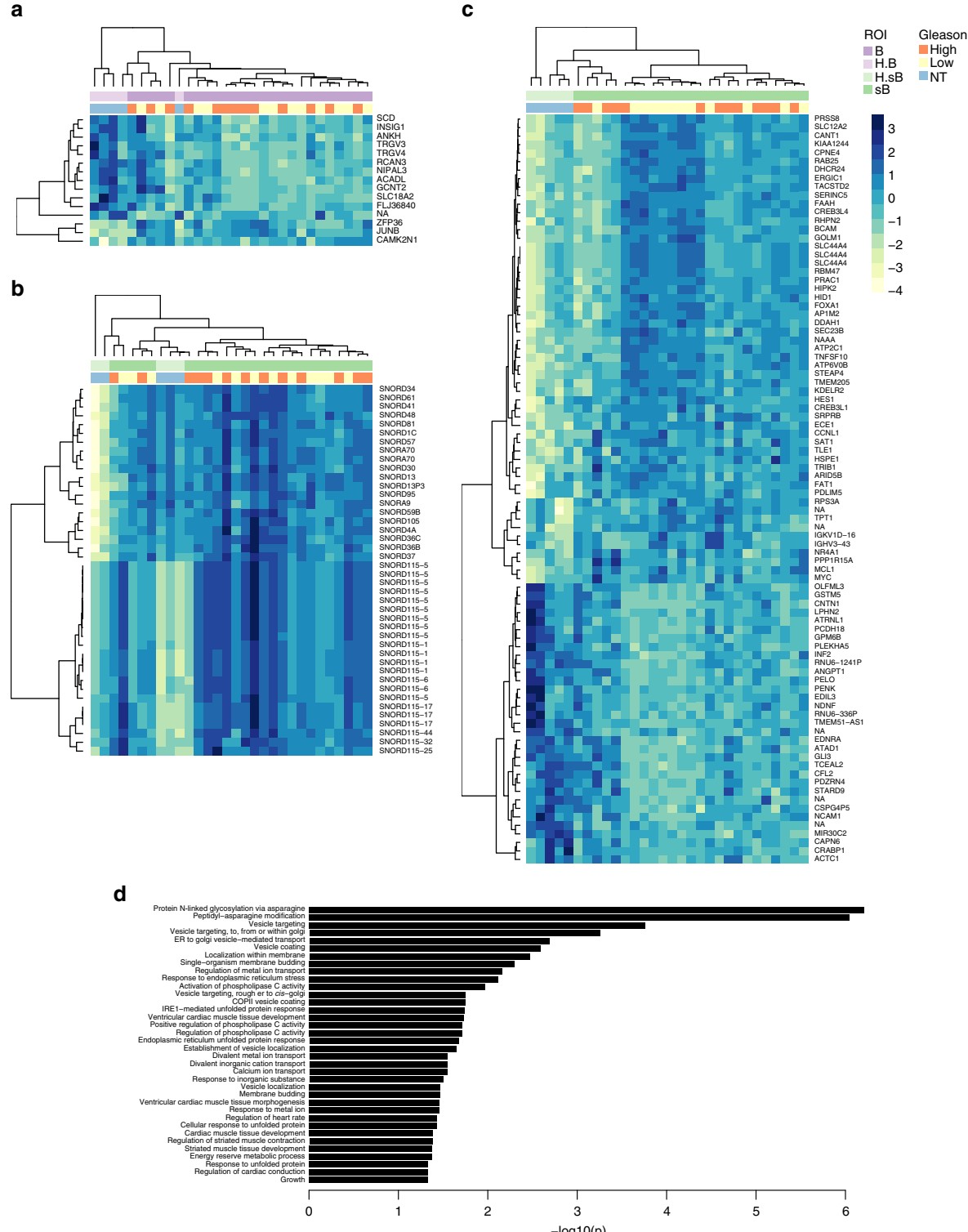

**Fig. 3** Genes and pathways differentially expressed in benign tissue between cystoprostatectomy and RP specimens. **a** Heatmap of genes differentially expressed in benign epithelium of prostate cancer patients and cystoprostatectomy patients without prostate cancer. Gleason grade corresponds to the grade of the prostate tumor present in the block and NT (no tumor) denotes cystoprostatectomy cases. **b** Heatmap of genes and **c** heatmap of SNORDs differentially expressed in stroma surrounding benign glands from prostate cancer patients and cystoprostatectomy patients without prostate cancer. **d** –Log10 FDR-values from the pathways analysis of the genes differentially expressed in benign stroma using GO biological processes annotations

tumor-associated stroma by a single number in order to study behavior of this group of genes as a whole. When comparing the high and low-Gleason score cases, the difference in the score was highly statistically significant (Fig. 4a; t-test, $P \leq 10^{-3}$). Interestingly, while none of the individual genes from the signature

reached statistical significance when stroma surrounding benign glands in the prostates bearing high-Gleason and low-Gleason cancers was compared (sB.high-sB.low), the difference in ssGSEA scores for this comparison was marginally significant (Fig. 4b; t-test, $P = 0.08$).

**Table 2 Genes differentially expressed between stroma adjacent to high-Gleason grade and low-Gleason grade tumors in LCM data and publicly available studies used for validation**

| Affy ID | Gene symbol | LCM | | | TCGA | | GSE46691 | | | |
|---|---|---|---|---|---|---|---|---|---|---|
| | | | | | | | Gleason | | Outcome | |
| | | logFC | *P*-value | FDR | *P*-value | FDR | *P*-value | FDR | *P*-value | FDR |
| 8132557 | AEBP1 | 0.987 | $<10^{-4}$ | 0.011 | 0.004 | 0.015 | 0.009 | 0.019 | 0.024 | 0.041 |
| 8042439 | ANTXR1 | 0.841 | $<10^{-4}$ | 0.042 | 0.015 | 0.032 | 0.002 | 0.005 | 0.003 | 0.013 |
| 8170648 | BGN | 1.059 | $<10^{-4}$ | 0.012 | 0.009 | 0.024 | $<10^{-4}$ | $<10^{-4}$ | 0.004 | 0.015 |
| 7898793 | C1QA | 0.747 | $<10^{-4}$ | 0.007 | 0.115 | 0.132 | 0.159 | 0.190 | 0.087 | 0.105 |
| 7898805 | C1QB | 0.786 | $<10^{-4}$ | 0.012 | 0.180 | 0.197 | 0.084 | 0.106 | 0.046 | 0.065 |
| 7898799 | C1QC | 1.014 | $<10^{-4}$ | 0.012 | 0.103 | 0.131 | 0.002 | 0.005 | 0.001 | 0.007 |
| 7960744 | C1R | 0.865 | $<10^{-4}$ | 0.042 | 0.076 | 0.117 | 0.061 | 0.081 | 0.067 | 0.084 |
| 7953603 | C1S | 1.005 | $<10^{-4}$ | 0.026 | 0.099 | 0.131 | 0.029 | 0.044 | 0.017 | 0.032 |
| 8001800 | CDH11 | 0.711 | $<10^{-4}$ | 0.046 | 0.017 | 0.033 | $<10^{-4}$ | 0.002 | 0.002 | 0.010 |
| 8016646 | COL1A1 | 1.149 | $<10^{-4}$ | 0.012 | 0.001 | 0.003 | 0.001 | 0.003 | 0.006 | 0.015 |
| 8046922 | COL3A1 | 0.867 | $<10^{-4}$ | 0.043 | 0.004 | 0.015 | 0.001 | 0.003 | 0.012 | 0.024 |
| 7980908 | FBLN5 | 0.849 | $<10^{-4}$ | 0.026 | 0.047 | 0.084 | 0.058 | 0.081 | 0.033 | 0.050 |
| 7906767 | FCGR2B | 0.747 | $<10^{-4}$ | 0.002 | 0.082 | 0.118 | 0.352 | 0.384 | 0.056 | 0.075 |
| 8178811 | HLA-DRB1 | 0.973 | $<10^{-4}$ | 0.047 | 0.246 | 0.257 | 0.483 | 0.504 | 0.129 | 0.141 |
| 8180003 | HLA-DRB1 | 0.944 | $<10^{-4}$ | 0.043 | | | | | | |
| 7965403 | LUM | 1.274 | $<10^{-4}$ | 0.043 | 0.113 | 0.132 | 0.003 | 0.008 | 0.005 | 0.015 |
| 8129573 | MOXD1 | 0.607 | $<10^{-4}$ | 0.016 | 0.009 | 0.024 | 0.013 | 0.024 | 0.371 | 0.387 |
| 7908924 | PRELP | 0.788 | $<10^{-4}$ | 0.046 | 0.076 | 0.117 | 0.023 | 0.036 | 0.032 | 0.050 |
| 7977615 | RNASE1 | 0.714 | $<10^{-4}$ | 0.042 | 0.283 | 0.283 | 0.278 | 0.318 | 0.126 | 0.141 |
| 8103254 | SFRP2 | 0.727 | $<10^{-4}$ | 0.008 | 0.001 | 0.003 | $<10^{-4}$ | 0.001 | 0.009 | 0.022 |
| 8139087 | SFRP4 | 1.074 | $<10^{-4}$ | 0.002 | $<10^{-4}$ | 0.001 | $<10^{-4}$ | $<10^{-4}$ | 0.001 | 0.007 |
| 8146863 | SULF1 | 0.721 | $<10^{-4}$ | 0.012 | 0.011 | 0.024 | $<10^{-4}$ | $<10^{-4}$ | 0.001 | 0.007 |
| 8130867 | THBS2 | 0.678 | $<10^{-4}$ | 0.012 | $<10^{-4}$ | 0.001 | 0.005 | 0.010 | 0.001 | 0.007 |
| 7952268 | THY1 | 0.589 | $<10^{-4}$ | 0.043 | 0.006 | 0.019 | 0.015 | 0.025 | 0.012 | 0.024 |
| 8101774 | TMSB4X | 0.657 | $<10^{-4}$ | 0.042 | NA | NA | 0.728 | 0.728 | 0.817 | 0.817 |
| 8067007 | TMSB4X | 0.664 | $<10^{-4}$ | 0.043 | | | | | | |
| 8166072 | TMSB4X | 0.675 | $<10^{-4}$ | 0.049 | | | | | | |

*FDR* false discovery rate, *LCM* laser capture microdissection

**Validation in external data**. Next the stromal signature (Table 2) was applied to the prostate data from The Cancer Genome Atlas[20]. In a comprehensive re-review of these 333 tumor samples by a group of GU pathologists a large variation in tumor purity was reported. Specimens with low purity contain a lot of stroma, which made them good candidates for preliminary validation of this stromal signature associated with Gleason grade. Cases were grouped into those with relatively high stromal content (tumor cellularity ≤ 40%) and cases enriched for tumor epithelium (tumor cellularity ≥ 80%). ssGSEA score of the stromal gene signature was calculated. A significant difference of the ssGSEA signature score between 3 + 3 and 8 + Gleason in both low tumor cellularity (Fig. 4c; *t*-test, $P = 0.006$) and high tumor cellularity (Fig. 4d; *t*-test, $P = 0.02$) subsets was found, but the difference was smaller and less significant in high cellularity samples, despite the larger sample size. This demonstrates that while it is possible to observe signal from the stromal genes in specimens with relatively low stromal content, it might be significantly diluted. Therefore, it is important to interpret prostate expression data as a function of stromal content.

Similarly, the signature was applied to stromally enriched samples (see Methods section) from the publicly available gene expression data from the Mayo clinic cohort[37] (GEO accession number GSE46691). The signature was significantly different between high-Gleason grade and low-Gleason grade samples (*t*-test, $P < 2\times10^{-10}$), and between cases that did or did not develop metastasis (Fig. 4d). The AUC of the signature score alone for predicting metastatic events using logistic regression model was 0.67, and together with Gleason score 0.74. In this cohort, Gleason score alone predicted outcome with AUC of 0.70.

**Validation using immunohistochemistry**. Protein expression of selected genes in the signature was tested by immunohistochemistry (IHC) to verify cell of origin. Only genes with IHC-validated antibodies were tested. As examples, the only significant gene in the epithelial compartment, ALCAM was overexpressed in the epithelial component of Gleason 8 tumors (Fig. 4f, g), and the stromal gene SULF1 was highly expressed in stroma adjacent to high-grade, but not low-grade tumor (Fig. 4h, i).

**Discussion**

The traditional consensus is that tumorigenesis is caused by mutations exclusive to epithelial cells that promote increased growth and invasive capacity, eventually resulting in metastasis. For some time, compelling data primarily derived from pre-clinical models have suggested that the microenvironment within which the cancer cells reside plays a pivotal role in cancer initiation and progression. Further, altered microenvironment may even precede genetic alterations in epithelial cells. Our results show that changes in the microenvironment are important contributors to tumor initiation and may affect progression.

We observed that stromal, but not the epithelial gene expression, obtained from benign areas (away from invasive tumors) in RP specimen differs significantly from that of prostates without cancer. Pathways such as N-glycosylation and the unfolded protein response (UPR) were upregulated in RP benign stroma compared to cystoprostatectomy specimens. These pathways are important in a variety of biological processes such as nutrient sensing or control of lipogenesis and are commonly altered in cancer. For instance, UPR can be an androgen

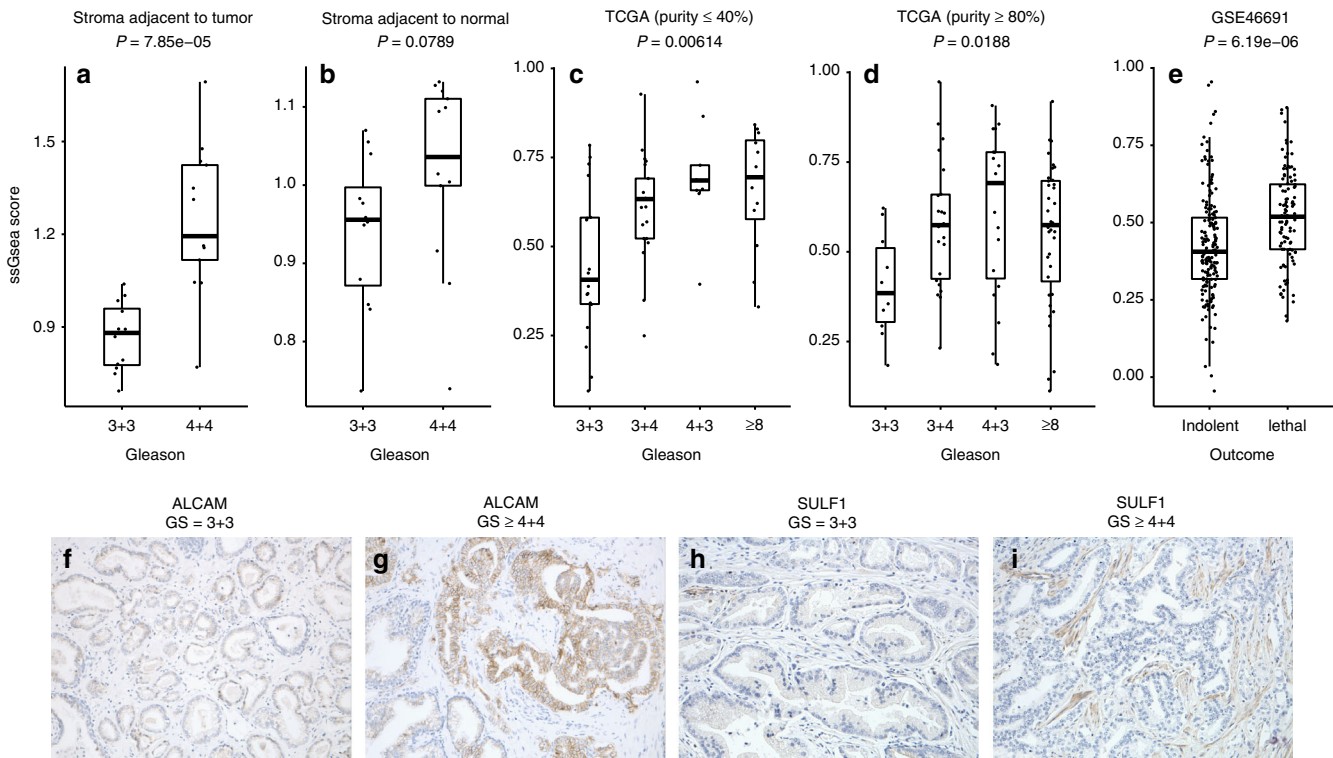

**Fig. 4** Genes differentially expressed in stroma surrounding high-Gleason grade and low-Gleason grade prostate tumors. **a** ssGSEA score of the stromal signature in tumor adjacent stroma; **b** ssGSEA score of the stromal signature in benign-adjacent stroma; **c** ssGSEA score of the stromal signature in low cellularity TCGA samples; **d** ssGSEA score of the stromal signature in high cellularity TCGA samples; **e** ssGSEA score of the stromal signature between indolent and lethal cases from GSE46691 cohort; **f** Immunohistochemical staining of epithelial ALCAM in Gleason 3 + 3 case; **g** Immunohistochemical staining of epithelial gene ALCAM in Gleason 4 + 4 case; **h** Immunohistochemical staining of stromal SULF1 in Gleason 3 + 3 case; **i** Immunohistochemical staining of stromal SULF1 in Gleason 3 + 3 case. *P*-values in (**a**–**e**) are from corresponding *t*-tests

responsive process in prostate cells and an aberrant UPR can lead to suppression of apoptosis, increased protein expression, and survival of prostate cancer cells. Metabolic challenges such as fluctuations in nutrient availability, hypoxia and increased demand on protein synthesis, can lead to perturbation of endoplasmic reticulum (ER) function, accumulation of misfolded proteins, and ER stress. In an attempt to restore ER homeostasis, the cell mounts a response called the UPR, a set of intracellular signaling pathways that aim to adjust the protein folding capacity of the cell[38]. Translational control of protein synthesis is therefore important for prostate cancer cell proliferation and survival but the role of stromal cells in this regard is novel, perhaps suggesting that a stromal environment exists in some individuals that is permissive for survival and proliferation of transformed epithelial cells.

Gleason grade is one of the strongest clinical predictors of prostate cancer progression and outcomes. An mRNA signature associated with Gleason grade improves risk stratification[23]. We identified only one gene differentially expressed between high-grade and low-grade tumor epithelium, *ALCAM*, a TGFβ responsive gene, previously shown to be associated with metastasis[35]. It is well known that TGF beta signaling plays important regulatory roles in stromal-epithelial interactions in both prostate development and tumorigenesis. Differences between gene expression in stroma adjacent to high-grade and low-grade cancer were much more striking: 25 genes were differentially expressed. All genes comprising this stromal signature of Gleason were more highly expressed in stroma from high-Gleason cases than those from low grade. The fact that gene expression from stroma across Gleason grades is more different

than that in the epithelial tumor compartment confirms the importance of the microenvironment and suggests that more work to develop drugs that specifically target the stroma is warranted.

Interestingly, among the 24 stromal genes differentially expressed across high and low grade were genes expressed by the immune system including complement, as well as many genes that are expressed in osteoblasts and osteoblast-like cells. The complement cascade is known to be an effector arm of innate immunity, playing a role in clearance of pathogens as well as in tumor immune surveillance. The complement system also plays a role in cartilage and bone development, as well as in regenerative pathways in injured tissue (reviewed in ref. [39]). Of note, some complement proteins are distributed throughout immature, developing bone and appear to be important in osteogenesis. Uncontrolled complement activation can also promote inflammation. Consistent with these findings, bone remodeling pathways were upregulated predominantly in stroma adjacent to malignant epithelium, and to a much lesser extent in benign or PIN adjacent stroma. The stromal genes lumican (*LUM*), *COL1A1* and *BGN*, belonging to both the signature we report here and comprising all stromal genes in the commercial Onco-typeDx kit[40], are also interesting in terms of the theme of bone remodeling. *COL1A1* is an osteoblastic differentiation marker[41] and *BGN* modulates angiogenesis and bone formation during fracture healing[42]. As prostate cancer most commonly metastasizes to bone, and Gleason 8 tumors are more likely to metastasize than Gleason 6, the finding of the overexpression of bone remodeling pathway in high-grade stroma is particularly interesting. The interaction of prostate cancer with the bone

microenvironment contributes to self-perpetuating progression of cancer in bone and the osteoclast-targeted agents zoledronic acid and denosumab decrease metastases to bone in metastatic castration-resistant prostate cancer[43]. We speculate that this prostate stromal environment may prepare cells from high-grade tumors to thrive in bone.

We successfully validated the association of the stromal Gleason signature with Gleason score in TCGA data. Not surprisingly, the signature was more strongly associated with Gleason in tumors with lower purity that have a higher percentage of stromal tissue. The signature was also significantly associated with lethal disease in expression data from the Mayo clinic cohort, although its prognostic power is likely to be suboptimal in this patient data set because the Mayo clinic data was designed to be enriched for epithelium. As the analysis of TCGA data suggests, we expect to observe stronger performance of the signature for prostate cancer prognosis in the stroma enriched specimens. A recently published study[44] that utilized patient derived xenograft models to develop a stromal signature of metastatic potential corroborates our findings that gene expression in stroma is predictive of prostate cancer outcome.

Interestingly, this Gleason signature was also borderline significantly different in stroma from benign areas of the prostates with high-grade and low-grade tumors. Additionally, when examining all gene expression data, we observed that benign stroma from men with high-grade tumors was more similar to cystoprostatectomy stroma than low-grade benign stroma, despite the fact that in our samples benign stroma from high-grade cases was physically closer to a tumor focus than in low-grade cases. This could suggest that there is a "prostate-wide" difference in the stroma of men who develop low-grade disease that allows for the development of well differentiated cancer with low malignant potential. Additional larger scale studies with benign stroma from healthy individuals and prostate cancer patients' samples taken repeatedly at different distances from tumor foci are needed to validate these findings. However, if confirmed, this would provide convincing evidence that it might be possible to identify a prognostic signature from stroma from biopsies that do not contain malignant epithelial cells. In prostate cancer, negative biopsies are a common occurrence and a significant clinical problem that results from random sampling in a PSA screened population. After a man has had an elevated PSA, but a negative biopsy, the normal stroma could be used to determine if he seems at risk only for low-grade disease or at risk for aggressive disease. This could help determine if and when he should return for a follow-up biopsy. In addition, a stromal signature in biopsies without neoplastic tissue may be of importance in the context of active surveillance.

While, we focused on comparing patients with Gleason scores 6 and 8+, many men are diagnosed with Gleason score 7 disease. The data from our current study does not permit us to comment on how the stroma behaves in these patients, but from the Mayo cohort data and TCGA data, it appears that the stromal signature in Gleason score 7 tumors falls in between Gleason score 6 and 8, suggesting an intermediate state of Gleason 7 stroma. A further investigation of the stroma in Gleason score 7 cases is necessary to clarify the gene expression profiles of stroma adjacent to pattern 3 and pattern 4 within Gleason 7.

This study comprehensively assesses gene expression from microdissected prostate tissue specimens, focusing on epithelial and stromal compartments across progression. Despite the moderate sample size used for discovery and scarcity of the external gene expression data from specimens with high stromal content, we were able to validate our key findings in the publicly available data.

## Methods

**Clinical specimens**. A total of 135 prostate cancer patients who underwent radical prostatectomy have been recruited from the following Institutions: Harvard School of Public Health, Boston, USA; Guy's and St Thomas NHS Foundation Trust/King's College London, UK; Prostate Cancer Research Consortium, Ireland; Orebro University Hospital, Sweden; S.Orsola-Malpighi Hospital Bologna, Italy. All patients provided informed written consent approved by each local IRB and research ethics committees.

Pathologists selected prostate cancer cases with Gleason grade 6 and Gleason grade ≥8. Cases with Gleason 7 were excluded from the study. Cases were selected according to the presence of sufficient amount of PIN, prostate cancer and normal prostate tissue in the same block/slide. In addition, cases with minimal inflammation were chosen. Immunohistochemistry for CD45 and CD163 were performed to assess the lymphocytic and macrophage infiltrate, respectively (Supplementary Fig. 4). After case review 25 cases were finally selected for the study. Thirty prostates from cystoprostatectomy cases were collected from patients with bladder cancer to be included in the study as normal controls for prostate. Cystoprostatectomy patients were not treated with BCG. Among the cystoprostatectomies the cases with incidental prostate cancer or excessive inflammation in the stromal component or atrophy in the epithelial component were excluded after pathology review and 5 out of 30 cases were selected for microdissection.

**Digital-pathology**. Slide digitalization and circling of the regions of interest (ROI) was centralized. The slides selected according to the above criteria were scanned with an Aperio CS2 instrument and put on a dedicated proprietary website protected by regulated access. Separate circling of the epithelial and stromal components in cancer, PIN and normal tissue areas was performed on digitalized H&Es using the Aperio ImageScope software V.10.35.1800 (representative examples of circling are shown in Supplementary Fig. 5). Annotated pathology scans were remotely accessed for the laser capture microdissection.

**Laser capture microdissection and gene expression profiling**. Following ROI review by digital annotation, LCM was performed on the Arcturus platform (Life Technologies), overnight incubation in lysis buffer/Proteinase K and subsequent RNA extraction by AllPrep (Qiagen) and quantification by RiboGreen assay (Life Technology). The differences between the means of stromal areas captured by LCM between Gleason 6 and Gleason 8 cases were not statistically significant (two-sample $t$-test $P$-values were 0.41, 0.2, and 0.88 for sB, sP and sT areas, respectively). The images of all LCM caps with captured ROI areas are available upon request. To accommodate the low RNA concentration and yields associated with microdissected tissues, we utilized the SensationPlus FFPE method. Twenty nanograms of total RNA at a concentration of 2.5 ng/μl was used to measure RNA expression across the whole transcriptome on the Affymetrix Gene Array STA 1.0.

**Normalization and differential gene expression analysis**. Pre-processing of the microarray data consisted of adjusting raw data at the probe level for technical variables, such as batches, overall median of the fluorescence intensities in each array and fraction of the probes with intensity higher than background levels. Adjusted values were normalized using RMA (robust multichip average) method[45]. There were no extreme outliers or failing samples, therefore we retained all assayed samples and ROIs for further analysis. We used transcript clusters from the 'main' category with log-median intensity of three in at least one of the ROIs.

We used random effects linear models approach to account for correlations between compartments within cases using Bioconductor package limma[46]. We adjusted for multiple comparisons using the Benjamini-Hochberg false discovery rate (FDR) method. A FDR ≤0.05 was considered significant. Significantly differentially expressed genes with the fold-changes not exceeding 1.5 were not reported. For pathway analysis we used a Wilcoxon test implemented in geneSetTest function in limma, signed and unsigned moderated $t$-statistics from linear model fits were used to rank the genes. Overrepresentation analysis for Supplementary Data 2 was done using Fisher's exact test. Gene Ontology Biological Processes annotations were downloaded from MSigDb[47] (used for all analysis) and Enrichment Map Gene Sets collections (http://download.baderlab.org/EM_Genesets/, used for the analysis presented in Fig. 3d and Supplementary Data 2). For analysis we only considered gene sets with less than 200 and more than 20 genes. Benjamini-Hocheberg FDR method was used to correct for multiple comparisons.

To identify GO biological processes enriched in the epithelial (or stromal) compartment across progression for Fig. 1 we identified genes overexpressed in epithelium and in stroma in benign tissue from cystoprostatectomies (H), in benign tissue from RP (B), PIN form RP (P), and tumor from RP (T). Direction and strength of overexpression was identified from the sign and magnitude of the moderated $t$-statistic in the corresponding linear model contrasts: H.B-H.sB, B-sB, P-sP and T-sT. For subsequent for gene sets The genes were ranked accordingly to their overexpression in epithelial and stromal compartments for subsequent gene sets analysis.

**ssGSEA signature score.** ssGSEA scores were computed using GSVA Bioconductor package. The ssGSEA algorithm assigns an enrichment score for each sample, that characterizes joint up- or down- regulation of a set of genes (in our case the set of genes comprising a stromal signature) in relation to remaining reference genes measured in the sample. We used the genes that were significantly upregulated in sT vs T comparison as a reference set (logFC thresholding was not applied here) for computing the score. We used the ssGSEA signature score to compare joint overexpression of the genes that belong to the signature between different samples in the LCM gene expression data and in the publicly available data used for validation. Higher ssGSEA scores correspond to stronger joint upregulation of the signature genes. For the TCGA and GSE46697 data sets both signature genes and reference set were subsetted to the genes measured in each study.

**Selection of stromally enriched samples.** In order to identify stromally enriched GSE46691 samples, we computed ssGSEA scores of the genes found to be significantly different with negative logFC in T-sT comparison (~ 3000 genes) in our LCM data using all measured genes as reference set. The scores computed on this set of genes had high correlations 0.34 and 0.82 with 1-Tumor Cellularity values inferred by pathologist and RNA-Seq-based computational estimates in TCGA data. Stromally enriched samples were defined as those having score above the median of the distribution of the score across all samples.

**Immunohistochemistry.** Immunohistochemical staining of ALCAM and SULF1 was performed on 4 μm sections using the Bond Refine Detection System on the Leica Bond Rx automated immunostainer. The sections were automatically deparaffinized, antigen retrieval was done with Citrate buffer (pH 6.0) and processed for 20 min. The slides were incubated with the antibody against ALCAM (Rb polyclonal, HPA010926, Sigma-Aldrich) and SULF1 (Rabbit polyclonal, HPA054728, Sigma-Aldrich) at dilution of 1:200 for 60 min, respectively. Normal stomach tissue was used as positive controls for ALCAM, and smooth muscle tissue was used as positive controls for SULF1, respectively. Smooth muscle and liver were used for negative/weak control for ALCAM and SULF1, respectively. Omission of the primary antibody was utilized as a blank control.

For double immunohistochemistry staining, 4 μm FFPE sections were prepared, after deparaffinization, and antigen retrieval sections were incubated with primary antibodies, anti-CD45 (Rb monoclonal, #13917, Cell Signal Technology) at dilution of 1:250 and CD163 (Ms mAb, NCL-L-CD163, Leica Biosystems) at dilution of 1:200 for 30 min. Followed with MACH2 Double stain 2 antibodies (Mouse-HRP + Rabbit-AP, MRCT525, Biocare) for 30 min at room temperature. With Vulcan Fast Red (FR805, Biocare) and Betazoid DAB (BDB2004, Biocare) as Chromogens, CD45 signal is red and CD163 signals as brown, and then counterstained with hematoxylin. Tonsil was used as positive controls for CD45, placenta was used as positive controls for CD163; and colon and skeletal muscle were used for negative/weak controls.

**Data availability.** LCM gene expression data generated and analyzed in this study was deposited to the Gene Expression Omnibus (GEO) with accession number GSE97284. Annotations for 333 TCGA prostate cancer samples were downloaded from cBioPortal (http://www.cbioportal.org/study?id=prad_tcga_pub#summary) and corresponding RSEM normalized gene expression values from FireHose portal (http://firebrowse.org/?cohort=PRAD&download_dialog=true). Mayo clinic cohort data were downloaded from GEO, accession number GSE46691. All other remaining data are available within the Article and Supplementary Files, or available from the authors upon request.

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

## Acknowledgements

We thank Jane Hayward for help in preparing the figures. This work was supported by a Challenge award from the Prostate Cancer Foundation. M.L.'s work is also supported by NIH grants RO1CA131945, R01CA187918, DoD PC130716, P50 CA90381, and the Prostate Cancer Foundation. S.T. and G.P. were in part supported by R01CA174206. S.T. was also supported by P50 CA90381 and R21CA185787. The Irish Prostate Cancer Research Consortium bioresource is supported by the Irish Cancer Society (PCI11WAT) and Wellcome Trust-Health Research Board Dublin Centre for Clinical Research.

## Author contributions

M.L., L.A.M., N.E.M., M.J.S., and G.P. conceived the study M.V.H., O.A., S.-O.A., R.W.W., S.P., S.P.F., and L.A.M. contributed FFPE specimens for analysis A.B., R.L., M.F., F.G., and M.L. provided resources for pathology review and performed the review M.B., C.B., and C.Z. performed L.C.M., RNA extraction and gene expression profiling ST performed statistical analysis of the data M.L., S.T., M.B., K.L.P., L.A.M, G.P., and M.J.S. interpreted the results S.T., M.L., K.L.P., M.B. wrote the manuscript R.W.W., M.V.H., L.A.M., N.E.M., M.S., and M.F. critically read and edited the manuscript

## Additional information

**Competing interests:** An international patent application covering the findings described in the manuscript was filed by M.L., S.T., M.B., and L.A.M. on November 11, 2016 and assigned application number PCT/US2016/061519. The remaining authors declare no competing financial interests.

