## [Peer Review File · Nature Communications]

Reviewers' comments:

Reviewer #1 (Remarks to the Author):

It is well established that the stromal cells adjacent to the adenocarcinomas take on a reactive phenotype, commonly referred to as cancer-associated fibroblasts (CAFs). CAFs have long been shown to be associated with cancer initiation, progression as well as and metastasis. To better understand this process at the molecular level, the current study undertook gene profiling in different stroma cell populations adjacent to benign, PIN and cancerous regions from patient specimens as well as from stromal cells of normal prostates without cancer (cystoprostatectomy cases). In addition, the gene signatures of the benign and cancerous epithelium in the cancer patient specimens were interrogated and compared to the normal epithelium from cystoprostatectomy cases. Importantly, all samples from prostate cancer patients were well annotated and rigorously selected based on grade and sufficient sample material for proper analysis after laser capture. In total, 25 cancer patients were profiled, giving robust power to the present data set.

To this readers' surprise, the data show that although similar gene profiles were found in the normal epithelium of cyctoprostaectomy patients and the non-neoplastic (benign) epithelium of prostate cancer specimens, there were significant differences in stroma cells adjacent to these epithelial regions. Furthermore, when comparing tumor-adjacent stromal in the cancer specimens, there was an upregulation of bone remodeling and immune signatures in the high grade specimens versus the lower-grade specimens. Based on the present data sets, the present data suggests that an altered microenvironment (i.e. stroma) may precede genetic alterations in epithelial cells, which, although previous hinted at in preclinical models, has not been previously shown in patient samples. Thus the authors concluded that stroma microenvironment may influence prostate cancer initiation as well as metastatic progression.

Although the experimental design is very well conceived and rigorous and the ideas presented viz-a-viz the gene signatures are novel, additional data analysis are needed to support the current conclusion in the study.

Comments: Major revision

1. "Benign epithelium in prostates with and without tumor were similar in gene expression space, stroma away from tumor was significantly different from that in prostates without cancer". To support this conclusion, principal components analysis plots comparing H (cystoprostatectomy epithelium) vs B (benign epithelium from PCa cases) and sH (cystoprostatectomy stroma) vs sB (stroma adjacent to nenign epithelium in PCa cases) are needed to show separated clusters of sH and sB, but not H and B.
2. Principal components analysis plots in Fig S1A simply compared epithelium and stroma which undoubtedly showed a clear separation of clusters. In Fig S3 with comparison of H vs T, sH vs sT with high and low Gleason grade, there does not seem to be any clear separation of clusters in either epithelium or stroma as stated by authors.
3. "A stromal gene signature reflecting bone remodeling and immune-related pathways was upregulated in high compared to low Gleason grade cases". The authors should provide the GSEA enrichment plots for both pathways.
4. The heatmap in Fig 1A shows FDR values of B (benign), P (PIN) and T (tumor) as compared to H (cystoprostatectomy) epithelial samples. What are the FDR values of group H compared to? This has not been clarified.
5. Legend for Fig 3D is missing.

Reviewer #2 (Remarks to the Author):

The authors use LCM and gene expression analysis to study the prostate stromal gene expression signature compared to the epithelium in glands with and without cancer. They identified genes differentially expressed between high grade tumor epithelium and low grade tumor epithelium.

They report upregulated genes in the stroma of cancer that corresponded to macrophages and osteoblast-like cells, as well as wound healing, complement and hematopoietic bone marrow markers.

The authors apply the single sample gene enrichment score ssGSEA to compare the stroma in high and low grade Gleason Cases and find significant differences.

Originality: Investigators in the past (Gregg et al and Dakhova et al, referenced in the manuscript) have used LCM to study gene expression in human prostate cancer stroma and have made the same general conclusion that the stroma gene signature in cancer glands is different from the non cancer glands. The difference in the present paper is that the authors study a somewhat larger number (n=30) of patients compared to these previous studies (n=17), and the present study correlates the stromal gene expression with clinical grade. The study is valuable because it uses LCM to specifically isolate/enrich the stromal and epithelial populations.

1. The findings of stroma macrophage and osteoblast like gene markers are of interest. It would be valuable if the authors stained (IHC) the stroma of the cases for macrophage (CD68) and myeloblast markers to explore the cellular infiltrate in the stroma that correlates. The manuscript could have higher value for the field if they used this finding to propose and experimentally support a novel concept concerning how the stroma and the epithelium are "co-dependent" and "co-evolve" as claimed in the abstract.
2. The reason for choosing the ssGSEA is not discussed, particularly when other gene signature panels exist. It was unclear if the genes found in the ssGSEA were the same or different to the ones covered in the figures and tables.
3. The number of individual cases within each Gleason grade is relatively small and this reduces the power and the significance. Accordingly they report only two markers ALCAM (epithelium) and SULF1 (stroma) that are significant enough to verify by IHC staining. The number of each sub group is very small and statistical conclusions comparing small groups are not meaningful. Perhaps a different grouping strategy with more cases per group would be more revealing.
4. The total amount of stroma can vary considerably from one tumor/specimen to the next and in the past investigators have shown that the total volume of stroma is prognostic. The authors must control for this variable, and more details must be provided about how the LCM samples are normalized between cases.

We would like to thank the reviewers for their interest in our work and thoughtful comments. We revised our manuscript to clarify the details as requested by the reviewers (in italics). Our response to each comment is presented below. Edits to the main text of the manuscript are marked in red in the submitted revised version.

Reviewer #1

1. *“Benign epithelium in prostates with and without tumor were similar in gene expression space, stroma away from tumor was significantly different from that in prostates without cancer”. To support this conclusion, principal components analysis plots comparing H (cystoprostatectomy epithelium) vs B (benign epithelium from PCa cases) and sH (cystoprostatectomy stroma) vs sB (stroma adjacent to benign epithelium in PCa cases) are needed to show separated clusters of sH and sB, but not H and B.*

We agree, and presented the comparisons between benign epithelium from cystoprostatectomies and RP, and benign stroma from cystoprostatectomies and RP that the reviewer is requesting in the principal components plots in figure S3.

2. *Principal components analysis plots in Fig S1A simply compared epithelium and stroma which undoubtedly showed a clear separation of clusters. In Fig S3 with comparison of H vs T, sH vs sT with high and low Gleason grade, there does not seem to be any clear separation of clusters in either epithelium or stroma as stated by authors.*

Gene expression data is prone to batch effects and often in large studies (our study had 165 microarray samples) the main signal/variability in the data might not correspond to underlying biological differences, but rather be due to technical artifacts and noise. We therefore decided to include the principal components plot in Figure S1 as a proof-of-principle, showing that indeed our high dimensional data immediately reflects phenotypic differences rather than batch effects.

Figure S3 is actually a comparison of H vs B and sH vs sB (as described in the response to question 1). We compared the epithelial and stromal tissue from cystoprostatectomies with benign tissue from men with high Gleason grade and low Gleason grade tumors elsewhere in the prostate. We agree with the reviewer that separation between these groups was not very apparent in our original figure S3. We recalculated the principal components using centered and scaled data (originally, we did not scale the data prior to PCA) which reveals a much clearer pattern. We also removed the word “cluster” in the figure legend, because we did not mean that the pattern was found as a result of applying some clustering algorithm.

We also updated our figure legend to better clarify which regions of interest are depicted in the plot.

Old legend:

Supplementary Figure 3 Principal components plots for (A) benign epithelium and (B) benign stroma from cystoprostatectomy and RP cases. While gene expression seems to be

homogeneous for epithelial cells, expression in the stroma seems to cluster, with benign stroma from RPs with high grade tumors being intermediate between stroma of RPs with low grade tumors and cystoprostatectomy cases with no tumors.

New legend:

Supplementary Figure 3 Principal components plots for (A) benign epithelium and (B) benign stroma from cystoprostatectomy and RP cases. Color of the points reflect 3 groups of samples: NT - benign tissue from cystoprostatectomy patients that had no tumor, “high” and “low” – benign tissue from RP patients that had high and low Gleason grade disease respectively. While gene expression seems to be more homogeneous for epithelial cells, expression in the stroma seems to ~~cluster~~ be more structured, with benign stroma from RPs with high grade tumors being intermediate between stroma of RPs with low grade tumors and cystoprostatectomy cases with no tumors.

3. *“A stromal gene signature reflecting bone remodeling and immune-related pathways was upregulated in high compared to low Gleason grade cases”. The authors should provide the GSEA enrichment plots for both pathways.*

GSEA belongs to a family of so-called “**non-cutoff**” gene sets analysis methods, and therefore is designed to identify gene sets and pathways associated with certain phenotype by using all measured genes that are ranked according to the strength of their association with the phenotype, for e.g. t-statistic from differential expression tests. Because of this we could not apply GSEA methodology and associated with it enrichment plots to identify pathways associated with a cut-off based 27 probesets (24 unique genes) stromal signature. This signature was derived from differential expression analysis comparing stroma surrounding high and low grade tumors by selecting genes that were statistically significant at 0.05 FDR and with a fold change ≥ 1.5 . Instead, to statistically support our statement we performed overrepresentation analysis using Fisher’s exact test which is often used in such scenarios (reviewed in Ackermann M, Strimmer K: A general modular framework for gene set enrichment analysis. BMC Bioinformatics. 2009, 10: 47-10.1186/1471-2105-10-47.). We now present the results of overrepresentation analysis explicitly in supplementary table S2. The vast majority of the biological processes found to be significantly overrepresented in our stromal gene signature were immune related, though we also found SKELETAL SYSTEM DEVELOPMENT (GO:0001501) to be overrepresented.

4. *The heatmap in Fig 1A shows FDR values of B (benign), P (PIN) and T (tumor) as compared to H (cystoprostatectomy) epithelial samples. What are the FDR values of group H compared to? This has not been clarified.*

We apologize for the unclear legend to the figure. We updated the legend to clarify that columns of the heatmap show FDR values for the pathways enriched in H (benign tissue from cystoprostatectomies), B (benign tissue from RP specimens), P (PIN), and T (tumor) epithelium in panel A; and in H.sB (benign stromal tissue from cystoprostatectomies), sB (stroma adjacent to benign epithelium tissue from RP specimens), sP (stroma adjacent to PIN), and sT (stroma adjacent to tumor) stroma in

panel B. For, example to obtain the leftmost column marked as H (healthy individuals) we performed differential expression analysis between H.B and H.sB specimens, ranked genes according to their overexpression in healthy epithelium and performed gene sets analysis using that ranked list. FDRs from this gene sets analysis were plotted in leftmost column of panel A. Next, using the same H.B vs H.sB comparison we ranked genes according to their overexpression in stroma, performed gene sets analysis and FDRs from that analysis constituted H-column in panel (B). Similarly, we obtained FDRs for B, P and T columns using B vs sB, P vs sP and T vs sT comparisons respectively.

Old legend:

Figure 1: Heatmaps of the GO biological processes differentially enriched in (A) epithelial and (B) stromal compartments across stages of prostate cancer progression. H denotes comparisons obtained from cystoprostatectomy data. The cells in the heatmaps are colored according to the FDR of the process in the gene set analysis.

New legend:

Figure 1: Heatmaps of the GO biological processes enriched in (A) epithelial and (B) stromal compartments across healthy prostate tissues and stages of prostate cancer progression. The cells in the heatmaps are colored according to the FDR of the process in the gene set analysis.

We also included the following explanation in the Methods section:

To identify GO biological processes enriched in the epithelial (or stromal) compartment across progression for Figure 1 we identified genes overexpressed in epithelium and in stroma in benign tissue from cystoprostatectomies (H), in benign tissue from RP (B), PIN form RP (P), and tumor from RP (T). Direction and strength of overexpression was identified from the sign and magnitude of the moderated t-statistic in the corresponding linear model contrasts: H.B-H.sB, B-sB, P-sP and T-sT. The genes were ranked accordingly to their overexpression in epithelial and stromal compartments for subsequent gene sets analysis.

5. Legend for Fig 3D is missing.

We corrected the figure legend.

Reviewer #2

1. *The findings of stroma macrophage and osteoblast like gene markers are of interest. It would be valuable if the authors stained (IHC) the stroma of the cases for macrophage (CD68) and myeloblast markers to explore the cellular infiltrate in the stroma that correlates. The manuscript could have higher value for the field if they used this finding to propose and experimentally support a novel concept concerning how the stroma and the epithelium are “co-dependent” and “co-evolve” as claimed in the abstract.*

We thank the reviewer for this insightful comment. In fact, in the selection of cases for this work, we had made it a point to select only cases with minimal inflammatory infiltrates, in order to obtain a stromal signature derived from the purest population of mesenchymal cells. While we avoided visible lymphocytic infiltrates, macrophages are much harder to recognize on H&E. As a result, we have now stained all cases with CD45 (pan-lymphocytic marker) and CD163 (a macrophage marker more specific than CD68, used routinely in clinical specimen in our department). The data obtained confirm minimal infiltrates in all cases utilized for LCM. We have now added a Supplementary Figure 5 to demonstrate this. The same block used for LCM was used for immunohistochemical analysis. These results suggest that it is indeed the adjacent stroma and not the inflammatory infiltrate that “communicates” with the adjacent neoplastic epithelium and changes with the evolution from benign to malignant, and from low to high grade in invasive lesions.

2. The reason for choosing the ssGSEA is not discussed, particularly when other gene signature panels exist. It was unclear if the genes found in the ssGSEA were the same or different to the ones covered in the figures and tables.

While numerous gene signatures for general prostate cancer exist (for tumor vs normal, for example), the purpose of this ssGSEA was to test the association of a combined “signature” comprised of all of the genes that differentiated high grade from low grade disease. The 27 probesets corresponding to 24 unique genes were found by differential expression analysis, and not by any version of the gene sets analysis, and comprised our stromal signature. We then used ssGSEA score as tool to summarize expression of all of these signature genes into a single number per sample. Then the samples were compared using the assigned ssGSEA score. Both in the LCM data and in TCGA and Mayo clinic validation sets we observed that higher Gleason grade was associated with higher signature scores (Figure 4 A-D), which means that genes comprising the signature were jointly upregulated in high vs low grade disease, as expected.

We modified the ssGSEA subsection of the Methods section to clarify this as follows:

ssGSEA signature score

ssGSEA scores were computed using GSVA Bioconductor package. The ssGSEA algorithm assigns an enrichment score for each sample, that characterizes joint up- or down- regulation of a set of genes (in our case the set of genes comprising a stromal signature) in relation to remaining reference genes measured in the sample. We used the genes that were significantly upregulated in sT vs T comparison as a reference set (logFC thresholding was not applied here) for computing the score. We used the ssGSEA signature score to compare joint overexpression of the genes that belong to the signature between different samples in the LCM gene expression data and in the publicly available data used for validation. Higher ssGSEA scores correspond to stronger joint upregulation of the signature genes. For the TCGA and GSE46697 data sets both signature genes and reference set were subsetted to the genes measured in each study.

3. *The number of individual cases within each Gleason grade is relatively small and this reduces the power and the significance. Accordingly they report only two markers ALCAM (epithelium) and SULF1 (stroma) that are significant enough to verify by IHC staining. The number of each sub group is very small and statistical conclusions comparing small groups are not meaningful. Perhaps a different grouping strategy with more cases per group would be more revealing.*

We agree that statistically significant results in a study with moderate power might be biased and overestimate differential expression. We therefore validated our findings in two large independent data sets – TCGA and Mayo Clinic cohort (GSE46691). These results are presented in Table 2. Even though we selected stromally rich TCGA and Mayo clinic samples (please, see “Online methods” section) they contained sizable amounts of epithelium, that diluted the signal from the stroma (as evident from Figure 4C and 4D). Even in this unfavorable scenario we found that majority of the genes that were differentially expressed between stroma surrounding high and low grade tumors in the LCM data remained statistically significant in TCGA and Mayo clinic cohorts. We therefore believe that despite moderate sample size we obtained a reliable list of stromal genes associated with the differences in Gleason grade that validated in the independent data.

Fig. 1 . COL1A1 immunohistochemistry. Note stromal expression

ALCAM was chosen for IHC validation as the only epithelial gene found to be significant in comparisons of high vs low grade malignant epithelium. And SULF1 was shown as a representative example of a stromal marker. However, we did not limit our analysis to these markers. We confirmed the stromal origin and overexpression in high vs low grade cases of 2 more genes (COL1A1 and FBLN5) from the signature, by

IHC (e.g. see COL1A1 below expressed in stromal cells – Fig 1). We also checked location and level of expression of all genes for which IHC-grade antibodies were tested in The Protein Atlas

(<http://www.proteinatlas.org/>) and confirmed their stromal origin.

4. *The total amount of stroma can vary considerably from one tumor/specimen to the next and in the past investigators have shown that the total volume of stroma is prognostic. The authors must control for this variable, and more details must be provided about how the LCM samples are normalized between cases.*

We absolutely agree that total amounts of stroma vary, making it difficult to reproducibly study conventionally collected samples. If the difference relating to otherwise homogenous stroma was only due to its volume, it would manifest in admixed samples as a difference in expression levels of stromal genes, that needs to be corrected for, but pure microdissected samples would show equal gene expression levels. We therefore performed laser capture microdissection (LCM) to obtain pure cell populations and found

differences in gene expression. Equal amounts of RNA were used for each microarray, therefore no additional normalization for stromal content was necessary for pure cell data. The normalization procedure that we used is described in the methods section.

We feel that these revisions improved considerably the quality of the manuscript and we hope that this new version is suitable for publication in *Nature Communications*.

Reviewers' comments:

Reviewer #1 (Remarks to the Author):

The authors have thoroughly satisfied all previous requests for clarification of data with revisions to the text and presentation of additional data sets. There are no further concerns.

Reviewer #2 (Remarks to the Author):

The authors have substantially improved the manuscript. However I am still concerned about the authors response to the following reviewer comment quoted from the response letter "The total amount of stroma can vary considerably from one tumor/specimen to the next and in the past investigators have shown that the total volume of stroma is prognostic. The authors must control for this variable, and more details must be provided about how the LCM samples are normalized between cases.

We absolutely agree that total amounts of stroma vary, making it difficult to reproducibly study conventionally collected samples. If the difference relating to otherwise homogenous stroma was only due to its volume, it would manifest in admixed samples as a difference in expression levels of stromal genes, that needs to be corrected for, but pure microdissected samples would show equal gene expression levels. We therefore performed laser capture microdissection (LCM) to obtain pure cell populations and found

Fig. 1 . COL1A1 immunohistochemistry. Note stromal expression

differences in gene expression. Equal amounts of RNA were used for each microarray, therefore no additional normalization for stromal content was necessary for pure cell data. The normalization procedure that we used is described in the methods section."

Comment: For the reader it is unclear how the cases are normalized between cases. If one tumor has a larger percent component of stroma, or a higher capture efficiency by the LCM, then the total volume of stroma will be different from one case to the next. This will mean that the total RNA will be higher if there is more total stroma captured from a given cases (e.g. number of LCM shots, number of cell equivalents, or area of stroma). The total RNA species for a given marker may therefore be elevated simply based on the volume or efficiency of the LCM procurement. This is particularly true for low abundance species that may not reach the lower limit of detection for a lower volume of stroma even though the relative expression rate per cell or total volume is the same. An internal control or LCM measurement must be used and this must be explained clearly for the reader.

Reviewers comment:

Comment: For the reader it is unclear how the cases are normalized between cases. If one tumor has a larger percent component of stroma, or a higher capture efficiency by the LCM, then the total volume of stroma will be different from one case to the next. This will mean that the total RNA will be higher if there is more total stroma captured from a given cases (e.g. number of LCM shots, number of cell equivalents, or area of stroma). The total RNA species for a given marker may therefore be elevated simply based on the volume or efficiency of the LCM procurement. This is particularly true for low abundance species that may not reach the lower limit of detection for a lower volume of stroma even though the relative expression rate per cell or total volume is the same. An internal control or LCM measurement must be used and this must be explained clearly for the reader.

Response:

We had stated before that we agree with the concern on stromal content. At the request of the reviewer, we hereby compared the total amounts of stroma for each case in the laser capture microdissected slides obtained from the LCM experiment. We compared the means of stromal areas between Gleason 6s and Gleason 8s and found no significant difference between the two (Fig 1). We added this information into the Methods section. Consequently, the amount of RNA obtained from each case, when all LCM caps for each ROI were pooled, did not significantly differ by Gleason grade in the stromal or epithelial compartments. The images of caps for all ROIs and samples will be made available to the readers upon request, the images may be accessed by the reviewer using temporary Dropbox link:

[REDACTED]

Moreover, as stated in the methods section identical amounts of material from each sample and ROI, namely 20 ng of total RNA at a concentration of 2.5ng/ul, were submitted for the microarray analyses.

Figure 1: Amount of stromal area captured by LCM process

In addition, the Dana-Farber microarray core that performed Affymetrix assays runs internal control samples with generic RNA identical between batches. There were two control samples run with our specimens. As evident from the quality control boxplots neither raw nor normalized data exhibit any patterns associated with the differences between regions of interest or control samples (Fig 2). Variability in the raw data is consistent with our prior experience with the partially degraded formalin fixed paraffin embedded specimens. Distributions of the expressions after normalization are practically identical for all samples, as desired in a microarray experiment. Normalization procedure is described in the Method section in a subsection entitled **“Normalization and Differential Gene Expression Analysis.”** Both raw and normalized data were submitted to GEO and will be made public upon publication of the manuscript. The reviewers may access the data using a temporary link:
 [REDACTED]

Figure 2: Distributions of raw (top panel) and normalized (bottom panel) gene expression values across samples do not show patterns associated with the ROIs.

We would like to emphasize, however, that the target of this paper was to obtain a pure signature from each compartment and not to compare total stromal content with progression or with predictive parameters. In order to accomplish this goal, we had to apply the signature to much larger database TCGA. In the TCGA cohort tumor purity, inversely correlated to stromal content, was measured. When the stromal signature we derived was tested in this cohort, a significant difference was found between 3+3 and 8+ Gleason in both low ($p=0.006$) and high tumor cellularity ($p=0.02$). Critically and supporting this reviewer’s contention, the difference was smaller and less significant in high cellularity samples, despite the larger sample size. Thus, while the signal from the stromal genes in specimens with relatively low stromal content is detectable,

determining stromal content (e.g. by image analysis or perhaps with a novel method of normalization to housekeeping stromal genes) in specimens that are not microdissected becomes essential, as the reviewer suggests.

REVIEWERS' COMMENTS:

Reviewer #2 (Remarks to the Author):

The authors have done an excellent job of responding to the remaining review comments regarding normalization. The revisions are highly satisfactory.

REVIEWERS' COMMENTS:

Reviewer #2 (Remarks to the Author):

The authors have done an excellent job of responding to the remaining review comments regarding normalization. The revisions are highly satisfactory.

The reviewer comments did not contain any requests and do not require response.